# The Cutaneous leishmaniasis impact questionnaire: Translation, cross-cultural adaptation and validation in adults with Cutaneous leishmaniasis in Ethiopia

Derese Bekele Daba [1]*, Feleke Tilahun Zewdu[1,2,3], Yematawork Kebede Aragaw[1], Debisa Eshatu Wendimu[1], Abebaw Yeshambel Alemu[3], Mosisa Bekele Degefa[1], Nebiyu Sherefa Mohammed[1], Yohannes Hailemichael[1], Teklu Cherkose[1], Fikregabrail Aberra Kassa[1], Tedros Nigusse Ferede[1], Galana Mamo Ayana[1], Kidist Weldekidan Desta[1], Shimelis Nigusie Doni[4], Amel Beshir Mohammed[5], Fewzia Shikur Mohammed[5], Saba Maria Lambert[1,6], Sagni Chali Jira[1], Iris Mosweu[7], Catherine Pitt[7], Michael Marks[6], Stephen L. Walker[6‡], Endalamaw Gadisa[1‡], SHARP Collaboration[¶]

1 Malaria and Neglected Tropical Disease Research Division, Armauer Hansen Research Institute, Addis Ababa, Ethiopia, 2 Dermatology Department, Boru Meda General Hospital, Dessie, Amhara, Ethiopia, 3 Department of Epidemiology and Biostatistics, Institute of Public Health, College of Medicine and Health Sciences, University of Gondar, Gondar, Ethiopia, 4 ALERT Comprehensive Specialized Hospital, Addis Ababa, Ethiopia, 5 Department of Dermatovenereology, College of Health Sciences, Addis Ababa University, Addis Ababa, Ethiopia, 6 Department of Clinical Research, Faculty of Infectious and Tropical Diseases, London School of Hygiene and Tropical Medicine, London, United Kingdom, 7 Department of Global Health and Development, London School of Hygiene and Tropical Medicine, London, United Kingdom

¶ SHARP Collaboration is provided in the Acknowledgments.
‡ These authors are joint senior authors on this work.
* derestbekele@gmail.com

## Abstract

### Background

Cutaneous leishmaniasis (CL) is a vector-borne, neglected tropical disease of the skin. It is a public health problem in Ethiopia, associated with reduced health-related quality of life (HRQoL). The Cutaneous Leishmaniasis Impact Questionnaire (CLIQ), a CL-specific measurement, was developed and validated in Brazil. This study aimed to translate, culturally adapt, and validate the CLIQ in Amharic.

### Methods

Translation, cultural adaptation, and pilot-testing of an Amharic version of the CLIQ were performed, involving a group of experts and affected individuals. The translated Amharic version of the CLIQ was administered to adults with confirmed active CL between February and September 2023. The Amharic version of the CLIQ was evaluated using Cronbach's α, inter-rater reliability, and assessments of face, content, construct, and criterion validity.

**Data availability statement:** All relevant data are within the manuscript and its Supporting information files.

**Funding:** This study was funded by the Research and Innovation for Global Health Transformation (RIGHT) Programme [Grant Reference Number NIHR200125] of the National Institute for Health and Care Research (NIHR) https://www.nihr.ac.uk/. The fund was received by SW. The funder had no role in study design, data collection and analysis, decision to publish, or preparation of the manuscript.

**Competing interests:** The authors have declared that no competing interests exist.

## Results

The translated and culturally modified Amharic version of the CLIQ was administered to 250 adults with CL. Of these, 158 (63.2%) participants had localized CL, and 114 (45.6%) were categorized as having moderately severe CL at enrolment. The Amharic version of the CLIQ had acceptable internal consistency ($\alpha = 0.913$) and very good stability (ICC: 0.935 (95% C.I.: 0.908, 0.957)). It exhibited acceptable content validity with a modified kappa coefficient of 0.33 to 1.0. Confirmatory Factor Analysis revealed a two-cluster tool with factor loading of 0.33–0.83 for cluster 1 and 0.19 to 0.7 for cluster 2. A statistically significant difference was observed in median scores of severities ($P < 0.001$) and clinical phenotypes ($P = 0.009$). There was a significant reduction in CLIQ scores at Day 90 compared to Day 1 ($P < 0.05$). The clinically important difference of the CLIQ was calculated to be 12.

## Conclusion

The Amharic version of the CLIQ is a reliable and valid instrument to measure the HRQoL associated with CL in adults in Ethiopia and can be used as a patient-reported outcome measure in the assessment of CL and its treatment.

## Author summary

Cutaneous leishmaniasis is a skin disease that can cause long-lasting wounds, pain, and emotional distress. To better understand how the disease affects people's daily lives, we tested an Amharic version of a questionnaire called the Cutaneous Leishmaniasis Impact Questionnaire (CLIQ). A total of 250 people with the disease took part in this study. Most participants were young adults, and many had the common form of disease that affects the skin. Doctors rated their illness as mild, moderate, or severe. After 90 days of treatment, more than half of the participants showed clear improvement, and some were completely cured. We examined how well the questionnaire measured people's experiences about general impact as well as health service and treatment. The tool showed strong reliability, meaning it gives consistent results when used at different times. It also showed good ability to measure what it is meant to measure, such as the general impact of the disease and people's feelings about treatment and health services. The questionnaire was also able to show differences in impact between mild, moderate, and severe cases. Scores improved after treatment among those who felt better, showing that the tool can track meaningful changes in health. Overall, the Amharic CLIQ is a useful and trustworthy tool for understanding how cutaneous leishmaniasis affects people in Ethiopia

## Introduction

Cutaneous leishmaniasis (CL) is a neglected tropical disease of the skin caused by *Leishmania* species transmitted following the bite of female sandflies [1,2]. The World Health Organization (WHO) estimates that globally, nearly one million new cases of CL occur annually in 89 CL-endemic countries [3]. In Ethiopia, 50,000 new cases of CL are estimated to occur [3,4], of which Amhara, Tigray, Oromia, Southern Nations, Nationalities, and People's regions are most affected [3,5,6].

CL lesions most commonly affect exposed parts of the body, such as the face and limbs [1]. Lesions heal with scarring and can result in permanent damage to important anatomical structures, especially on the face, which are stigmatizing [7], affect mental health [8], self-esteem, and economic attainment [9]. CL is classified into localized CL (LCL), the most common form, mucocutaneous leishmaniasis (MCL), and diffuse CL (DCL) [10].

Adults in Ethiopia with active CL experienced reduced health-related quality of life (HRQoL), assessed using the Dermatology Life Quality Index (DLQI) [11]. CL affects HRQoL across psychosocial [7,12], mental, physical, and economic domains [9,13]. The Cutaneous Leishmaniasis Impact Questionnaire (CLIQ) is a disease-specific patient-reported outcome measure (PROM) intended to assess the impact of CL on affected individuals [14]. It was developed in Brazil and published in Portuguese and English. A reliable, validated Amharic version of the CLIQ will facilitate assessment of the impact of CL on affected individuals and a robust research tool [15] for the assessment of clinical outcomes in Ethiopia. We aimed to translate and culturally adapt the CLIQ into Amharic and validate it in adults with CL in Ethiopia.

## Methods

### Ethics statement

Ethical approval was obtained from the Ethics Committee of the London School of Hygiene and Tropical Medicine with reference number [26421], the ALERT/AHRI Ethical Review Committee [PO/23/21], and the Ethiopia National Research Ethics Review Committee [7/2–506/m259/35]. Written informed consent was obtained from participants after explanation on the purpose of the study. Confidentiality and privacy of participants were maintained throughout the study.

### Study design, setting and period

The study was conducted in two dermatology referral hospitals: ALERT comprehensive specialized hospital in Addis Ababa and Borumeda General Hospital in South Wollo zone, Northeast Ethiopia, between February and September 2023. The study was nested in a large CL cohort study as part of the multi-disciplinary Skin Health Africa Research Programme (SHARP). The CL cohort study protocol is described elsewhere [10].

### Instrument

The CLIQ is a recently developed patient-reported outcome measure used by clinicians and researchers to measure the impact of CL [16,17]. The CLIQ consists of 25 items categorized in seven domains namely Global impact, Physical/functional impact, social impact, Occupational impact, Economic impact, treatment and health service satisfaction, that constitute two clusters: Cluster 1 general impact (17 items), and Cluster 2 perception about the treatment and health services (HSS) impact (8 items). Each of the 25 items is scored from zero to four. Total CLIQ scores are calculated by adding item scores, giving a maximum score of hundred and a minimum score of zero. Higher scores indicate greater impact.

The DLQI is a ten-item general skin disease tool measuring HRQoL associated with "skin problems," first developed and validated in 1994. The DLQI has been translated and validated in more than 80 countries [18], including Ethiopia [19]. The DLQI contains ten questions in six domains namily symptom and feelings, daily activities, leiasure, job and schools, personal relationship and treatment. The 10 questions measure the impact of the dermatological condition over the past week, with one of four responses ranging from "not at all" to "very much" giving a total score ranging from zero to 30. Higher scores indicating worse HRQoL.

## Translation and cultural adaptation

The English version of the CLIQ [14] was translated independently into Amharic by two experts: a dermatologist who is a native Amharic speaker with a good understanding of English and a sociologist fluent in Amharic and English. A third bilingual individual reviewed the two translations, checking for inconsistencies or altered meaning of phrases or sentences.

Back translation to English of the Amharic versions was performed independently by an additional pair of experts to check consistency with the original English version. An expert six-member translation committee, consisting of two dermatologists, two public health experts, and two sociologists, checked, verified, and approved the translation to produce an initial Amharic version of the CLIQ, which was then pilot tested.

## Face validity

To assess the face validity of the CLIQ, five people with CL were interviewed to assess their comprehension of each item and to ascertain whether the items were conceptually difficult, unclear, embarrassing (not acceptable), or contained difficult words to understand. Modifications were agreed by the translation committee based on the interview results.

The modified version was then discussed with five different individuals with CL. A pre-final draft version was then made by the translation committee after the incorporation of comments and feedback [5,17].

## Content validity

The relevance of each of the 25 items in the pre-final draft version of the CLIQ was assessed by six experts, two each from the disciplines of public health, dermatology, and nursing. They graded the relevance of items using a Likert scale, with the responses "not relevant," "somewhat relevant," "quite relevant," and "highly relevant.". Item-related content validity index (I-CVI) and average scale-level content validity index (S-CVI/Ave) were calculated based on their responses [20,21].

## Study participants

Adult (above 18 years of age) Amharic speakers with laboratory-confirmed CL were enrolled consecutively in the study. An individual was diagnosed with CL when *Leishmania* infection was confirmed by microscopy or culture or the presence of *Leishmania* DNA in affected tissue. CL was classified as LCL, MCL, or DCL using study definitions [10].

## Sample size

A sample size of 10 participants per item is considered acceptable for validation studies [22]. We aimed to recruit 250 individuals.

## Data collection tools, procedures, and analysis

Socio-demographic and clinical data, including lesion characteristics, CL phenotype, physician-determined severity at enrolment were recorded using study definitions on standardized forms [10]. Physician Global Assessment was recorded at Day 90 [23].

Anonymized data were entered into REDCap version 13.10.1 (Powered by Vanderbilt). Data cleaning was performed to check for accuracy, completeness, consistency, and missing values. Data were analysed using STATA version 17. Descriptive statistics were presented using frequencies and percentages.

## Validity assessment

Confirmatory Factor Analysis (CFA) was used to show the factorial structure or construct of the CLIQ. Global model fitness indices (ratio of chi-square to degree of freedom ($\chi2/df$) < 5.0, root mean square error of approximation (RMSEA) ≤0.08,

comparative fit index (CFI) > 0.9, Tucker Lewis Index (TLI) > 0.9, and p > 0.05 for the chi-square test) were checked while reporting factor loading during CFA. Cronbach's alpha and intra-class correlation coefficients were calculated.

Known-group construct validity was assessed with physician-determined severity of CL (mild, moderate, or severe) and clinical phenotype: LCL, MCL, and DCL [20,23]. Median rank sum scores of each of the clusters were recorded using the Kruskal-Walli's rank sum test among the different CL phenotypes to check known-group construct validity.

CLIQ scores at enrolment were correlated with scores of the Amharic version of the DLQI to assess association [24]. Comparisons between CLIQ and DLQI were performed using Spearman's roh correlation coefficient and scatter plot. The Spearman correlation coefficient $(\rho) \geq 0.4$ was considered an acceptable correlation coefficient.

Responsiveness to change of the CLIQ following treatment was assessed by comparing CLIQ scores at Day 1 and Day 90 in individuals classified using Physician Global Assessment as worse/no change, minor improvement (1–50% of the lesion flattened), substantial improvement (51–99% flattened or repepithelialized) and cure (complete flattening or re-epithelialisation of the lesion) [23,24]. A Wilcoxon-signed rank test was used to compare the median difference of scores before initiation of treatment (Day 1) and at Day 90 (date of assessment of response to treatment).

Minimum Clinical Important Difference (MCID) and clinically important difference (CID) were determined using anchor-based and distribution-based approaches. The MCID and CID were determined using an anchor-based approach of participant reported change at day 90 assessed as "better," "somewhat better," and "somewhat worse." The MCID was calculated using the difference in mean score of individuals who were "somewhat better" at Day 90 and the CID using those who were "better" at Day 90.

Furthermore, medium effect size, ½ standard deviation (SD), and standard error of the mean (SEM) were used to detect CID using a distribution-based approach.

### Reliability assessment

The internal consistency of the CLIQ was assessed using Cronbach's alpha [20,25]. The inter-rater reliability was assessed by having two interviewers independently completing the CLIQ with the same participant on the same day, at least 60 minutes apart. We calculated the intra-class correlation coefficient (ICC) to assess agreement of responses [20,25]. The ICC score interpretation was: < 0.5 poor, 0.5–0.75 moderate, 0.75–0.9 good and >0.9 excellent.

## Result

### Translation and cultural adaptation

During translation, an expert committee (n = 6) ensured there were no contextual changes to each of the items, instructions, and response options. Both translators translated the term "leishmaniasis" directly to the Amharic term as "የቆዳ ሊሽማኒያሲስ," which is not a commonly used word, so the term "ቁንጭር/kunchir", the Amharic vernacular term for CL, was added. The response options were refined to better capture the intensity of responses by incorporating "how often" or "how much" across all items. Each item was scored on a 0–4 scale, resulting in a total possible score ranging from 0 to 100 for respondents who completed the questionnaire. Higher scores indicate a greater impact of CL.

### Face validity

During pilot testing with CL-affected individuals and review by the experts, various comments and suggestions were received for modifications. Key changes included adding definitions in brackets, particularly for terms in item 16, such as clothes (ልብስ), neckwear (ሻርፕ, የአንገት ልብስ), spectacles (መነፅር), hats (ኮፍያ), face masks (ማስክ), to clarify possible ways to cover wounds. Additionally, for item 17, terms like wedding ceremony (ሰርግ), memorial service (ተስካር), and funeral ceremony (ሞት,ቅሬ) were added to provide context for social activities that might be affected by CL. In response to the feedback from affected individuals and experts, the options "Not Applicable" or "I Don't know" were added as possible responses for each item, as some questions were not relevant to the interviewees. These responses are scored zero.

**Content Validity**

The S-CVI/Ave based on I-CVI was 0.9, a value above the recommended (≥0.78), and the S-CVI based on proportion relevance was 0.9, as well as S-CVI/UA was 0.68. The calculated modified kappa coefficient (k*) showed that most items had perfect agreement of experts, but 4 items (item numbers 8, 14, and 15) had moderate expert agreement, while item number 23 showed slight acceptance by experts (S1 Table).

**Validity and reliability assessment**

**Participant characteristics.** The final draft of the Amharic version of the CLIQ was completed by 250 participants with CL described in Table 1. The median age was 30 years (IQR: 22–45). The majority were in the 18–27-year age group. Most participants, 152 (60.8%) were male. 142 (56.8%) were recruited at Borumeda General Hospital. 142 of 250 (56.8%) were inpatients. The majority, 158 (63.2%), had LCL, 87 (34.8%) had MCL, and 5 (2.0%) had DCL.

Based on the treating dermatologist's assessment, 91 (36.4%) had mild disease, 114 (45.6%) moderate disease, and 45 (18.0%) severe disease. Furthermore, at Day 90, 64 (55.7%) showed substantial improvement and 33 (28.7%) had been cured.

**Reliability assessment.** The CLIQ scores ranged from 0 to 83, with a median score of 40.5 ((IQR)= 25–56). Items within Cluster 1 showed item-total correlations between 0.484 and 0.741, while those in Cluster 2 ranged from 0.096 to 0.338.

The Cronbach alpha for the Amharic version of the CLIQ was 0.913, showing acceptable internal consistency. The general impact scale and perception about the treatment and HSS had Cronbach alphas of 0.932 and 0.415, respectively (**Table 2**).

The stability of scores was evaluated in 59 participants using intraclass correlation coefficients (ICC). The cluster scores showed excellent stability with ICC values 0.957 (0.940, 0.971) for Cluster 1, 0.729 (0.542, 0.839) for Cluster 2, and the total scores had an ICC of 0.935 (95% C.I.: 0.908, 0.957), indicating very good reliability (P-value< 0.001).

**Validity assessment. Construct validity:** A confirmatory factor analysis was conducted to assess the factor structure of the CLIQ tool. The hypothesized two-factor/cluster model showed an acceptable fit (CFI = 0.912, TLI = 0.901, RMSEA = 0.06, SRMR = 0.08). Factor loadings ranged from 0.33 to 0.83 for Cluster 1 and 0.19 to 0.7 for Cluster 2, all statistically significant at p < 0.001. Construct reliability was adequate for all factors (CR > 0.70), and AVE values indicated good convergent validity (AVE > 0.50) for cluster I and low convergent validity for cluster 2. Discriminant validity was supported, with latent variable correlations below 0.80. The AVEs for each factor were greater than the squared correlations between the two clusters, supporting discriminant validity. Additionally, factor correlations were all below 0.85, indicating clear distinctions between constructs (S2 Table).

Convergent validity was assessed using spearman roh correlation coefficient ($\rho$). A significant correlation was observed between total CLIQ score and general impact ($\rho$ = 0.98) and perception on treatment and service satisfaction ($\rho$ = 0.45) (p < 0.01). Further analysis indicated a significant correlation between item-cluster total that ranged from 0.547 to 0.79, P = 0.01 for cluster I and 0.318 to 0.540, P = 0.01 for cluster 2, insuring convergent validity.

Spearman's correlation coefficient revealed that the overall scores of CLIQ and DLQI exhibited acceptable but low correlation (roh = 0.510) and a linear but weak correlation on the scatter plot ($R^2$ = 0.259).

**Known group validity:** The Mann-Whitney U-test indicated that the median score of CLIQ among LCL and MCL types of the lesion was significantly different (P = 0.02). Furthermore, the Kruskal-Wallis test also indicated that the median score of CLIQ was significantly different among mild, moderate, and severe types of lesions (p < 0.001). Pair-wise post-hoc comparison also exhibited a significant difference in median score among mild and moderate (p < 0.01), mild and severe (p < 0.01), and moderate and severe (p = 0.037) types of the lesions.

**Table 1. Socio-demographic and clinical characteristics of participants with active cutaneous leishmaniasis.**

| | | Frequency (%) [n = 250] | CLIQ score: median (IQR) |
|---|---|---|---|
| Age group (in years) | | | |
| | 18–27 | 103 (41.2) | 38 (24–53) |
| | 28–37 | 55 (22.0) | 43 (26–55) |
| | 38–47 | 41 (16.4) | 42 (24–63) |
| | 48–57 | 30 (12.0) | 39 (30–56) |
| | 58–67 | 14 (5.6) | 46.5 (35–58) |
| | 68–77 | 7 (2.8) | 31 (17–58) |
| Sex | | | |
| | Male | 152 (60.8) | 40 (25–55.5) |
| | Female | 98 (39.2) | 42 (25–56) |
| Educational status | | | |
| | No formal education | 62 (24.8) | 48 (30–59) |
| | Primary | 77 (30.8) | 38 (25–56) |
| | Secondary | 71 (28.4) | 41 (24–57) |
| | Post-secondary | 40 (16.0) | 35 (18–46) |
| Department attended | | | |
| | In-patient | 142 (56.8) | 43 (25–57) |
| | Out-patient | 108 (43.2) | 37.5 (37.5–53.5) |
| Number of Lesions | | | |
| | Single | 200 (80.0) | 43 (24–67) |
| | Multiple | 50 (20.0) | 48 (59–25) |
| Size of the largest lesion | | | |
| | Less than 40mm | 114 (45.6) | 43 (30–47) |
| | 40 mm or more | 136 (54.4) | 46.5 (27–58) |
| Clinical phenotype | | | |
| | LCL | 158 (63.2) | 35 (21–52) |
| | MCL | 87 (34.8) | 45 (32–59) |
| | DCL | 5 (2.0) | 58 (47–62) |
| Clinical Severity | | | |
| | Mild | 91 (36.4) | 28 (18–45) |
| | Moderate | 114 (45.6) | 43.5 (31–56) |
| | Severe | 45 (18.0) | 57 (38–66) |
| Global physician assessment at day 90 (N = 115) | | | |
| | Worse or no change | 2 (1.7) | 29.5 (12–47) |
| | Minor improvement | 16 (13.9) | 35.5 (22.5– 53) |
| | Substantial improvement | 64 (55.7) | 32.5 (15–42) |
| | Cured | 33 (28.7) | 30 (18–46) |
| Participant Opinion at Day 90(N = 115) | | | |
| | Better | 90 (78.3) | 31.5 (13–44) |
| | Somewhat better | 22 (19.1) | 37 (23–50) |
| | Somewhat worse | 3 (2.6) | 22 (12–47) |

Note: IQR: Inter quartile range; LCL: Localized Cutaneous Leishmaniasis; MCL: Mucocutaneous leishmaniasis; DCL: Diffuse Cutaneous Leishmaniasis

**Table 2. Item-level scores of CLIQ among participants with active cutaneous leishmaniasis (n = 250).**

| Clusters | Items | Median | Mean ± SD | Item-total correlation | Cronbach's alpha if item Deleted |
|---|---|---|---|---|---|
| Cluster 1 | G1 | 2 | 2.27 ± 1.48 | 0.667 | 0.907 |
| | PF4 | 1 | 1.38 ± 1.58 | 0.625 | 0.908 |
| | PF5 | 2 | 1.90 ± 1.20 | 0.732 | 0.906 |
| | PF6 | 2 | 1.40 ± 1.60 | 0.730 | 0.906 |
| | PF7 | 2 | 1.44 ± 1.86 | 0.603 | 0.908 |
| | E8 | 2 | 1.78 ± 1.48 | 0.741 | 0.905 |
| | E9 | 2.5 | 1.99 ± 1.38 | 0.680 | 0.907 |
| | E10 | 0 | 2.30 ± 1.58 | 0.484 | 0.911 |
| | E11 | 2 | 1.54 ± 1.46 | 0.563 | 0.909 |
| | O12 | 2 | 1.73 ± 1.63 | 0.715 | 0.906 |
| | O13 | 2 | 1.73 ± 1.56 | 0.665 | 0.907 |
| | Ec14 | 2 | 1.73 ± 1.63 | 0.728 | 0.906 |
| | Ec15 | 2 | 2.03 ± 1.56 | 0.651 | 0.907 |
| | Ec16 | 2 | 1.22 ± 1.49 | 0.515 | 0.910 |
| | S17 | 0 | 1.62 ± 1.46 | 0.521 | 0.910 |
| | S18 | 1 | 1.44 ± 1.52 | 0.577 | 0.909 |
| | S19 | 1 | 1.99 ± 1.61 | 0.671 | 0.907 |
| Cluster 2 | TI20 | 0 | 1.54 ± 1.66 | 0.377 | .913 |
| | TI21 | 1 | 2.16 ± 1.66 | 0.141 | .916 |
| | TI22 | 2 | 3.62 ± .89 | 0.108 | .915 |
| | HSS23 | 2 | 0.68 ± 0.92 | 0.096 | .915 |
| | HSS24 | 2 | 0.78 ± 1.12 | 0.338 | .913 |
| | HSS25 | 2 | 2.65 ± 1.30 | 0.117 | .918 |
| | HSS26 | 4 | 1.40 ± 1.52 | 0.172 | .915 |
| | HSS27 | 3 | 2.39 ± 1.89 | 0.282 | .914 |

Note: G- Global, PF- Physical functioning, E-Emotional, O-Occupational, Ec- Economic, S- Social, TI- Treatment impact, HSS- Health service satisfaction.

**Responsiveness to change:** Data for 115 participants were available at Day 90. The mean CLIQ score at enrolment for all participants was 41.54 ± 20.90 and the median 44.0 (IQR = 25–57). At Day 90 the mean for all participants was 31.43 ± 17.51 and median 32.0 (IQR = 15–46). There was a significant difference in CLIQ scores after treatment (Day 90) compared with enrolment for those categorized as having minor improvement, substantial improvement, or those assessed as cured (P-value<0.05). (**Fig 1**). Individuals categorised as worse or no change did not have a significant difference between their CLIQ scores at enrolment and Day 90.

Participant anchor-based assessment of CID and MCID revealed that the mean change for those who reported themselves to "better" be -11.5 ± 23.9. Participants who felt "somewhat better" had a mean change in CLIQ score of -4.8 ± 19.5. However, there was a greater improvement in CLIQ scores, -7.3 ± 28.2, for the three participants who felt they were "somewhat worse" at Day 90 compared to enrolment. Using a distribution-based approach, the CID was 10.46 with a medium effect size (0.5), 10.45 with ½ SD, and 6.17 with SEM.

## Discussion

We have translated and validated an Amharic version of the CLIQ. It demonstrated strong internal consistency and excellent stability and compares favourably with the original version [26,27]. Our study confirmed the two-cluster structure

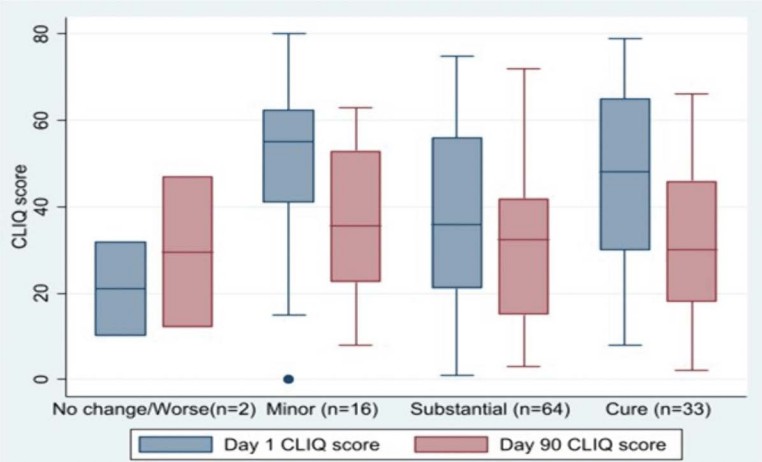

**Fig 1. Change in CLIQ scores at Day 1 and Day 90 with respect to Physician Global Assessment categorized as worse or no change, minor improvement, substantial improvement, and cured.**

(general impact and perception about treatment and HSS) of the CLIQ ([26]. We have shown the Amharic version of the CLIQ differentiates between physician-determined severities and is responsive to change following treatment.

Our Amharic version of the CLIQ exhibited robust psychometric properties, with high internal consistency observed for the total scale and general impact subscale consistent with the validation studies of the CLIQ in Brazil [26,28]. The subscale assessing perceptions of treatment and health services showed low internal consistency which was the case with the original CLIQ [28]. The significant differences in CLIQ scores following treatment in our cohort was demonstrated in Brazil [28]. The significant differences in CLIQ scores between individuals with LCL and MCL has not been demonstrated previously. The difference in CLIQ scores between our physician determined severity assessments, suggests a potential role for the CLIQ in categorising disease severity.

The Amharic version of the CLIQ is weakly correlated with the Amharic version of the DLQI, suggesting that the two tools are measuring different aspects of HRQoL associated with CL. The original version of the CLIQ correlated moderately well with the EQ-5D-3L [27]. This demonstrates that multiple tools may need to be used to accurately capture the HRQoL associated with CL. The use of multiple PROMs has been advocated for individuals with myeloma [29].

We propose that the Amharic version of the CLIQ has a CID of 12 based on the anchor-based approach using participants' assessments and supported by the distribution-based methods, which suggested a CID of 11.

We are unable to recommend a MCID for the Amharic version of the CLIQ due to the smaller improvement in CLIQ score of those who were "somewhat better" compared to those who were "somewhat worse" This is likely due to the small number of participants in the latter category.

The limitations of our study include its conduct in leishmaniasis referral centres, a small pre-test sample that may not fully represent individuals with CL, and subjective nature of clinician-assessed severity. Additionally, the small numbers of individuals categorized as "worse or no change" at Day 90, as well as those reporting themselves as "somewhat worse", prevented an analysis of MCID. Further research is needed to establish MCID of the Amharic version of the CLIQ. Participant self-assessment responses may have been subject to reporting bias.

Future work could develop adaptations of the CLIQ suitable for children and adolescents although in Brazil it has been used in individuals 15 years and older [27,28]. In rural Ethiopia, a recent household survey revealed that 56% of individuals with active CL were children 12 years old or younger [30].

This study contributes to the growing body of validated HRQoL tools for CL and provides a culturally adapted health instrument for use in Ethiopia [30,31]. Demonstrating the validity of the CLIQ in settings other than Brazil where it was developed lends credibility to the tool and potential utility to the wider CL community. The Amharic version of the CLIQ CID could be used as an outcome measure in much needed clinical trials, addressing a current gap, as no PROMs were included in the proposed clinical trial methodologies for LCL [32]. Incorporating the CLIQ and other appropriate PROMs into such recommendations may encourage further validation of such instruments in other languages spoken in CL endemic regions.

## Conclusion

The Amharic version of CLIQ exhibited acceptable reliability and validity, making it an appropriate outcome measure for assessment of individuals with CL in Ethiopia.

## Supporting information

**S1 Table. Expert scores and agreement for CLIQ items.**
(DOCX)

**S2 Table. Confirmatory factor analysis results with factor loading, construct reliability and average variance extracted.**
(DOCX)

**S3 Table. Amharic version of the Cutaneous leishmaniasis impact questionnaire.**
(PDF)

## Acknowledgments

Skin Health Africa Research Programme (SHARP) is a multi-center, multi-sector, multidisciplinary collaborative research work between Armauer Hansen Research Institute, Addis Ababa University from Ethiopia, and the University of Ghana. Kumasi Centre for Collaborative Research in Tropical Medicine (KCCR) from Ghana and London School of Hygiene and Tropical Medicine from the UK. Experts from all the above institutions were contributing in one way or another. Therefore, the Authors would like to extend their deepest gratitude to all who contributed to the success of this project. We wish to thank the individuals and communities for their participation in the work of the Skin Health Africa Research Programme. We acknowledge our fellow leishmaniasis researchers, Professor Endi Lanza Galvão, Dr Ana Rabello and Dr Gláucia Cota, who developed the original CLIQ.

## Author contributions

**Conceptualization:** Saba Maria Lambert, Catherine Pitt, Michael Marks, Stephen L. Walker, Endalamaw Gadisa.

**Data curation:** Derese Bekele Daba, Feleke Tilahun Zewdu, Yematawork Kebede Argaw, Debisa Eshatu Wendimu, Mossisa Bekele Degefa, Nebiyu Sherefa Mohammed, Yohannes Hailemichael, Teklu Cherkose, Fikregabrail Aberra Kassa, Tedros Nigusse Nigusse, Kidist Weldekidan Desta, Shimelis Nigusie Doni, Amel Beshir Mohammed, Fewzia Shikur Mohammed, Iris Mosweu.

**Formal analysis:** Derese Bekele Daba, Michael Marks, Stephen L. Walker, Endalamaw Gadisa.

**Funding acquisition:** Saba Maria Lambert, Catherine Pitt, Michael Marks, Stephen L. Walker, Endalamaw Gadisa.

**Investigation:** Derese Bekele Daba, Debisa Eshatu Wendimu, Shimelis Nigusie Doni, Amel Beshir Mohammed, Fewzia Shikur Mohammed.

**Methodology:** Derese Bekele Daba, Feleke Tilahun Zewdu, Yematawork Kebede Argaw, Debisa Eshatu Wendimu, Yohannes Hailemichael, Shimelis Nigusie Doni, Fewzia Shikur Mohammed, Saba Maria Lambert, Michael Marks, Stephen L. Walker, Endalamaw Gadisa.

**Project administration:** Yohannes Hailemichael, Saba Maria Lambert, Michael Marks, Stephen L. Walker, Endalamaw Gadisa.

**Resources:** Yohannes Hailemichael, Michael Marks, Stephen L. Walker, Endalamaw Gadisa.

**Software:** Derese Bekele Daba, Fikregabrail Aberra Kassa, Galana Mamo Ayana.

**Supervision:** Debisa Eshatu Wendimu, Yohannes Hailemichael, Saba Maria Lambert.

**Validation:** Derese Bekele Daba, Debisa Eshatu Wendimu, Yohannes Hailemichael, Teklu Cherkose, Fikregabrail Aberra Kassa, Galana Mamo Ayana.

**Visualization:** Derese Bekele Daba, Yohannes Hailemichael, Fikregabrail Aberra Kassa.

**Writing – original draft:** Derese Bekele Daba.

**Writing – review & editing:** Derese Bekele Daba, Feleke Tilahun Zewdu, Yematawork Kebede Argaw, Debisa Eshatu Wendimu, Abebaw Yeshambel Alemu, Mossisa Bekele Degefa, Nebiyu Sherefa Mohammed, Yohannes Hailemichael, Teklu Cherkose, Fikregabrail Aberra Kassa, Tedros Nigusse Nigusse, Galana Mamo Ayana, Kidist Weldekidan Desta, Shimelis Nigusie Doni, Amel Beshir Mohammed, Fewzia Shikur Mohammed, Sagni Chali Jira, Iris Mosweu, Catherine Pitt, Michael Marks, Stephen L. Walker, Endalamaw Gadisa.

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
