## [Decision Letter · Decision Letter 0]

24 Aug 2025

The Cutaneous Leishmaniasis Impact Questionnaire: translation, cross-cultural adaptation and validation in adults with Cutaneous Leishmaniasis in Ethiopia

Dear Dr. Daba,

Thank you for submitting your manuscript to PLOS Neglected Tropical Diseases. After careful consideration, we feel that it has merit but does not fully meet PLOS Neglected Tropical Diseases's publication criteria as it currently stands. Therefore, we invite you to submit a revised version of the manuscript that addresses the points raised during the review process.

Please submit your revised manuscript within 60 days Oct 23 2025 11:59PM. If you will need more time than this to complete your revisions, please reply to this message or contact the journal office at plosntds@plos.org. Please include the following items when submitting your revised manuscript:

We look forward to receiving your revised manuscript.

Kind regards,

Mitali Chatterjee

Academic Editor

Abhay Satoskar

Section Editor

Shaden Kamhawi

co-Editor-in-Chief

Paul Brindley

co-Editor-in-Chief

**Journal Requirements:**

1) Please upload the main figure as a separate Figure file in .tif or .eps format. For more information about how to convert and format your figure files please see our guidelines:

2) We have noticed that you have uploaded Supporting Information files, but you have not included a complete list of legends. Please add a full list of legends for your Supporting Information file (CLIQ_Amharic version.pdf) after the references list. Please ensure that the supplementary tables are labeled correctly.

3) Please amend your detailed Financial Disclosure statement. This is published with the article. It must therefore be completed in full sentences and contain the exact wording you wish to be published.

2) If any authors received a salary from any of your funders, please state which authors and which funders.

4) Your current Financial Disclosure states, " The Skin Health Africa Research Programme (SHARP) is funded by the Research and Innovation for Global Health Transformation (RIGHT) Programme [Grant Reference Number NIHR200125] of the National Institute for Health and Care Research (NIHR) https://www.nihr.ac.uk/. The funder had no role in study design, data collection and analysis, decision to publish, or preparation of the manuscript. ".

However, your funding information on the submission form doesn't indicate any funds. Please ensure that the funders and grant numbers match between the Financial Disclosure field and the Funding Information tab in your submission form. Note that the funders must be provided in the same order in both places as well.

Note If the reviewer comments include a recommendation to cite specific previously published works, please review and evaluate these publications to determine whether they are relevant and should be cited. There is no requirement to cite these works unless the editor has indicated otherwise.

**Comments to the Authors:**

**Please note that one of the reviews is uploaded as an attachment.**

**Reviewers' Comments:**

Reviewer's Responses to Questions

**Key Review Criteria Required for Acceptance?**

**Methods**

-Are the objectives of the study clearly articulated with a clear testable hypothesis stated?

-Is the study design appropriate to address the stated objectives?

-Is the population clearly described and appropriate for the hypothesis being tested?

-Is the sample size sufficient to ensure adequate power to address the hypothesis being tested?

-Were correct statistical analysis used to support conclusions?

-Are there concerns about ethical or regulatory requirements being met?

Reviewer #1: Are the objectives of the study clearly articulated with a clear testable hypothesis stated?

Yes

-Is the study design appropriate to address the stated objectives?

yes

-Is the population clearly described and appropriate for the hypothesis being tested?

yes

-Is the sample size sufficient to ensure adequate power to address the hypothesis being tested?

Line 325: What is the rationale of having two nursing officers to perform content validity of CLIQ

250 sample size is adequate

-Were correct statistical analysis used to support conclusions?

yes

-Are there concerns about ethical or regulatory requirements being met?

no

Reviewer #2: (No Response)

Reviewer #3: 1. yes

2. yes

3. Yes

4. Yes

5. Yes

6. Yes

**Results**

-Does the analysis presented match the analysis plan?

-Are the results clearly and completely presented?

-Are the figures (Tables, Images) of sufficient quality for clarity?

Reviewer #1: -Does the analysis presented match the analysis plan?

yes

-Are the results clearly and completely presented?

1. S Table I is in ampharic and unable to understand

2. S Table 2 is in ampharic and unable to understand

3. Line 224: Face validity has to be done using more than 10 individuals. Usually 15 (5 from one location x3). N=5 in this study may be insufficient. But this was identified as a limitation by the authors.

4. line 252: Classification of Physician determined severity of CL (mild, moderate, or severe) has to have guidelines as without a guideline, the interpretation could be very subjective. However, they have used a previously published method.

5. line 256: No guidelines for “minor improvement”, “substantial improvement”, Could be very subjective.

6. Line 261: Inter-rater reliability has to be performed by 2 raters. Not one rater repeating after 60 minutes. That is Intra-rater reliability and that also needs a minimum of 24 hr gap to prevent recall memory.

-Are the figures (Tables, Images) of sufficient quality for clarity?

1. Suggest to add the standard cutoff values of ICC scores for the reader to interpret easily. This can be added as foot notes under Table 2.

2. Include meaning of abbreviations as foot notes in tables including in supplementary tables

3. Include the significant levels (P values) in Fig 1 between day 0 and Day 90.

Reviewer #2: (No Response)

Reviewer #3: 1.yes

2.No

3.No

**Conclusions**

-Are the conclusions supported by the data presented?

-Are the limitations of analysis clearly described?

-Do the authors discuss how these data can be helpful to advance our understanding of the topic under study?

-Is public health relevance addressed?

Reviewer #1: Are the conclusions supported by the data presented?

yes

-Are the limitations of analysis clearly described?

yes, but not in a very acceptable manner.

-Do the authors discuss how these data can be helpful to advance our understanding of the topic under study?

not in a satisfactory manner

-Is public health relevance addressed? limited

Reviewer #2: (No Response)

Reviewer #3: 1. Yes

2. No

3. yes

4. yes

**Editorial and Data Presentation Modifications?**

Reviewer #1: (No Response)

Reviewer #2: (No Response)

Reviewer #3: Recommend Minor revision

**Summary and General Comments**

Reviewer #1: Overall this study describes a very subjective assessment method. Need improvement by using definitions in future

i.e How to categorize mild, moderate severe in CL?

How to categorise, minor and moderate improvement etc.

Discussion needs improvement and needs to describe the application and the usefulness of CLIQ

CLIQ has already been validated by Galvão et al. in 2018 and the need to translate into a local language and validating it again is important to the local community.

I do not see the importance of ampharic version to the international community. Suggest to submit this paper to a local journal

Reviewer #2: This is a diligently conducted study. I wondered why the range of scores is so wide in each of the pre- and post treatment category. It is the experience with the original CLIQ and DLQI questionnaires as well?

Are any component of the Amharic CLIQ contributing to this noise?

How different are the “somewhat better” compared to those who were“somewhat worse", can CLIQ MCID actually be recommended in other languages?

I think the discussion could be a bit more meaty with comparison with CLIQ adaptations in other contexts

Reviewer #3: Materials and methods:

1. The names of certain statistical test and other important terms have been misspelled these should be corrected throughout the manuscript.

2. The Face Validity section mentions that a modification was made based on interview results but does not specify what modification was made.

3. In the Validity Assessment section, “Known-group construct validity was assessed with physician-determined severity of CL (mild, moderate, or severe) and clinical phenotype: LCL, MCL, and (20, 23).” The phrase ends abruptly with “and” and an incomplete list, missing “DCL.” Complete the list by adding “DCL”.

4. The term global physician assessment (GPA) is introduced in the "Socio-demographic and Clinical Data" section without defining it earlier in the text. The author should mention the definition of GPA in manuscript.

5. The 'Data collection tools, procedures, and analysis' section mentions global model fit indices (e.g., χ²/df < 5.0, RMSEA ≤ 0.08) but does not clarify whether these thresholds were met in this study. The authors should state whether the model met these criteria.

Results:

1. Throughout the results section of the main manuscript, the median IQR are presented as single value. However, the IQR should be reported as a range (Q1 to Q3). This needs to be clarified.

Table 1: A. The table reports data for “Global physician assessment at day 90 (N=115)” and “Participant Opinion at Day 90 (N=115),” indicating that only 115 of the 250 participants were assessed at Day 90. However, the text does not explain why only 115 participants were included.

B. “Number of Lesion” should be pluralized to “Number of Lesions,” and “Total size of the lesion” should be “Total Size of Lesions” for consistency with multiple lesions.

C. The table lists “Total size of the lesion” with a cutoff of “Less than 40 mm” and “40 mm or more,” but it’s unclear whether this refers to the diameter of a single lesion or the cumulative size of multiple lesions.

3. Reliability assessment

A. The Cronbach’s alpha for Cluster 2 (0.415) is notably low, which is below the commonly accepted threshold for adequate internal consistency (≥0.7). The authors should address the low Cronbach’s alpha value for Cluster 2.

B. The number of items in Cluster 1 and Cluster 2 are not specified in the main manuscript and should be provided for ease of interpretation.

C. The text does not specify whether the item-total correlations and Cronbach’s alpha values were calculated for the entire sample (n=250) or a subset, nor does it indicate the number of items in each cluster, which could affect interpretation of the alpha values.

4. Table 2

A. Ensure all numerical values in this column have the same precision (e.g., two decimal places). For instance, PF6 has "1.40±1.60", which is consistent, but visually confirm all others align (e.g., TI22: "3.62±.89" should be "3.62 ± 0.89" for uniformity)

B. Cluster 2 items HSS23 (0.096), TI21 (0.141), TI22 (0.108), HSS25 (0.117), and HSS26 (0.172) show low item-total correlations, suggesting weak construct alignment. The author should revise these items and further validate them to enhance the CLIQ tool’s convergent validity and utility.

PLOS authors have the option to publish the peer review history of their article (what does this mean? ). If published, this will include your full peer review and any attached files.

**Do you want your identity to be public for this peer review?** For information about this choice, including consent withdrawal, please see our Privacy Policy .

Reviewer #1: No

Reviewer #2: No

Reviewer #3: No

**Figure resubmission:**
---

## [Decision Letter · Decision Letter 1]

28 Jan 2026

Response to Reviewers
Revised Manuscript with Track Changes
Manuscript

Shaden Kamhawi

co-Editor-in-Chief

Paul Brindley

co-Editor-in-Chief

**Reviewers' comments:**

**Key Review Criteria Required for Acceptance?**

**Methods**

-Are the objectives of the study clearly articulated with a clear testable hypothesis stated?

-Is the study design appropriate to address the stated objectives?

-Is the population clearly described and appropriate for the hypothesis being tested?

-Is the sample size sufficient to ensure adequate power to address the hypothesis being tested?

-Were correct statistical analysis used to support conclusions?

-Are there concerns about ethical or regulatory requirements being met?

Reviewer #2: (No Response)

Reviewer #4: This paper clearly describes in detail the process to translate, culturally adapt and validate CLIQ scale to be used in Ethiopia.

The study methodology is complete. Just to clarify, kindly suggest to add few additional details:

1)Line 207: Please include what are CLIQ domains Scale

2)Line 210: Please clarify if "Higher scores indicate greater impact" means worst evaluation?

3)Line 214: Please include what are the DLQI domains Scale

To facilitate text flow and comprehension:

4) Line 252. Section: Validity Assessment: Include and describe subsections: Face validity, Content validity, Construct validity and Criteria Validity

5) Line 267. Section: Data collection tools, procedures, and analysis: kindly suggest organize the statistical analysis by Validity and Reliability evaluation

**Results**

-Does the analysis presented match the analysis plan?

-Are the results clearly and completely presented?

-Are the figures (Tables, Images) of sufficient quality for clarity?

Reviewer #2: (No Response)

Reviewer #4: Results are presented according with the analysis plan.

Following the previous recomendations, and to be consistent, the authors can decide if the results could be presented in the same order: Validity Assessment: Face, Content, Construct, Criteria Validity, and Reliability Assessment.

**Conclusions**

-Are the conclusions supported by the data presented?

-Are the limitations of analysis clearly described?

-Do the authors discuss how these data can be helpful to advance our understanding of the topic under study?

-Is public health relevance addressed?

Reviewer #2: (No Response)

Reviewer #4: The conclusions are clear, and study limitations are presented.

Will be good to explain more details why the CLIQ is not correlated with the DLQI scale. In addition, to explain why cluster 2 had less internal consistence and less AVE compared with cluster 1.

**Editorial and Data Presentation Modifications?**

Reviewer #2: (No Response)

Reviewer #4: Please, review the titles and labels of supplementary tables.

**Summary and General Comments**

Reviewer #2: (No Response)

Reviewer #4: This study is relevant to evaluate the Health-Related Quality of Life (HRQoL) of ´patients with cutaneous leishmaniasis in Ethiopia.

PLOS authors have the option to publish the peer review history of their article (what does this mean? ). If published, this will include your full peer review and any attached files.

**Do you want your identity to be public for this peer review?** For information about this choice, including consent withdrawal, please see our Privacy Policy .

Reviewer #2: No

Reviewer #4: No

**Figure resubmission:**

**Reproducibility:** To enhance the reproducibility of your results, we recommend that authors of applicable studies deposit laboratory protocols in protocols.io, where a protocol can be assigned its own identifier (DOI) such that it can be cited independently in the future. Additionally, PLOS ONE offers an option to publish peer-reviewed clinical study protocols. Read more information on sharing protocols at https://plos.org/protocols?utm_medium=editorial-email&utm_source=authorletters&utm_campaign=protocols

---

## [Editor Report · Decision Letter 2]

23 Feb 2026

Dear Mr Daba,

We are pleased to inform you that your manuscript 'The Cutaneous Leishmaniasis Impact Questionnaire: translation, cross-cultural adaptation and validation in adults with Cutaneous Leishmaniasis in Ethiopia' has been provisionally accepted for publication in PLOS Neglected Tropical Diseases.

Best regards,

Mitali Chatterjee

Academic Editor

Abhay Satoskar

Section Editor

Shaden Kamhawi

co-Editor-in-Chief

Paul Brindley

co-Editor-in-Chief

---

## [Editor Report · Acceptance letter]

Dear Mr Daba,

We are delighted to inform you that your manuscript, "The Cutaneous Leishmaniasis Impact Questionnaire: translation, cross-cultural adaptation and validation in adults with Cutaneous Leishmaniasis in Ethiopia," has been formally accepted for publication in PLOS Neglected Tropical Diseases.

Best regards,

Shaden Kamhawi

co-Editor-in-Chief

Paul Brindley

co-Editor-in-Chief
